# Clinically Defined Lymphogranuloma Venereum among US Veterans with Human Immunodeficiency Virus, 2016–2023

**DOI:** 10.3390/microorganisms12071327

**Published:** 2024-06-29

**Authors:** Gina Oda, Joyce Chung, Cynthia Lucero-Obusan, Mark Holodniy

**Affiliations:** 1Public Health National Program Office, Department of Veterans Affairs, Washington, DC 20420, USA; joyce.chung@va.gov (J.C.); cynthia.lucero@va.gov (C.L.-O.); mark.holodniy@va.gov (M.H.); 2Division of Infectious Diseases and Geographic Medicine, Stanford University, Stanford, CA 94305, USA

**Keywords:** lymphogranuloma venereum, chlamydia infection, epidemiology, Veterans, HIV

## Abstract

We applied lymphogranuloma venereum (LGV) clinical case criteria to a cohort of 1381 Veterans positive for HIV and *Chlamydia trachomatis* (CT) from 2016 from 2023 and analyzed variables to ascertain risk factors for LGV and factors associated with the use of standard treatment regimens. In total, 284/1381 (20.6%) met the criteria for LGV. A total of 179/284 (63%) were probable cases, and 105/284 (37%) were possible cases (those meeting clinical criteria but with concurrent sexually transmitted infections (STI) associated with LGV-like symptoms). None had confirmatory CT L1–L3 testing. A total of 230 LGV cases (81%) presented with proctitis, 71 (25%) with ulcers, and 57 (20.1%) with lymphadenopathy. In total, 66 (23.2%) patients had >1 symptom of LGV. A total of 43 (15%) LGV cases were hospitalized. Primary risk factors for LGV were male birth sex (*p* = 0.004), men who have sex with men (*p* < 0.001), and the presence of STIs other than gonorrhea or syphilis (*p* = 0.011). In total, 124/284 (43.7%) LGV cases received standard recommended treatment regimens. Probable cases were more likely to receive standard treatment than possible cases (*p* = 0.003). We report that 20.6% of CT cases met clinical criteria for LGV among HIV-infected Veterans and that less than half of cases received recommended treatment regimens, indicating that LGV is likely underestimated and inadequately treated among this US population.

## 1. Introduction

Lymphogranuloma venereum (LGV) is a sexually transmitted infection (STI) caused by the L1–L3 serovars of *Chlamydia trachomatis* (CT). These serovars cause a more systemic, invasive form of chlamydia infection, which results in a lymphoproliferative reaction manifesting in a variety of symptoms. In the early stages of infection, a painless genital/rectal/oral ulcer or papule may appear, and/or proctitis with rectal pain, discharge (bloody or mucopurulent), tenesmus, and constipation. The secondary stage may involve the development of inguinal, femoral, or anorectal lymphadenopathy and fever, as well as other systemic signs of infection such as body aches and malaise. If untreated, infection may progress to the third stage, which can include chronic proctitis, colorectal fistulas, and strictures, as well as genital lymphedema (elephantiasis) [1,2,3,4].

LGV is endemic in some regions of Africa, Asia, South America, and the Caribbean islands and was previously thought to occur rarely in industrialized nations [5,6]. Starting in 2003, with outbreaks occurring in Western Europe among men who have sex with men (MSM), LGV is thought to be increasing in incidence outside of its endemic region [4,5,7]. Outbreaks have been reported in North America and Australia as well as Europe, primarily among MSM who are coinfected with human immunodeficiency disease (HIV) [8,9,10,11]. In the US, recognition of LGV outbreaks and understanding of its epidemiology is hindered because (1) testing to confirm CT infection with serovars L1–L3 is not routinely performed due to limited commercially available testing in the US, and (2) while CT is a nationally notifiable disease, LGV as a separate entity is not [7,12]. Despite recent guidance by the Council for State and Territorial Epidemiologists (CSTE) allowing reporting jurisdictions to distinguish between LGV and non-LGV CT infections reported to the Centers for Disease Control and Prevention (CDC), few jurisdictions have implemented such specific reporting [13,14]. In those European countries that routinely utilize confirmatory testing for LGV and have robust LGV surveillance and reporting systems (e.g., the United Kingdom, France, and the Netherlands), the incidence of LGV cases has been shown to be steadily increasing since 2003 [8,15]. However, the availability of LGV diagnostic testing and case reporting varies across European countries. In a study by Cole et al., CT-positive specimens taken from patients in European Union (EU) countries without existing LGV reporting systems were tested for LGV using real-time polymerase chain reaction (RT-PCR), with overall positivity found to be 25.5% among MSM, representing significant underdiagnosis [16]. Similarly, a study by Chi et al. sampled remnant CT-positive rectal swab specimens from US state and local public health laboratories and found a 14% LGV positivity rate, indicating undetected circulation of LGV-associated serovars in the US [7].

Treatment of LGV requires an extended course of doxycycline 100 mg orally two times/day for 21 days or alternatively, azithromycin 1 g orally once weekly for 3 weeks or erythromycin base 500 mg orally four times/day for 21 days [1]. Because non-LGV CT infections are routinely treated for only 7 days of doxycycline (or alternative regimens such as levofloxacin 500 mg orally for 7 days or azithromycin 1 g orally in a single dose), accurate diagnosis of LGV based on presenting clinical syndrome is critical, lacking confirmatory diagnostic capability. Antimicrobial resistance is not considered a major factor in treating LGV; the goal of longer-duration treatment is to cure the infection and prevent irreversible tissue damage [1].

We conducted our study in the US Veterans Health Administration (VHA), which consists of over 1355 inpatient and outpatient healthcare facilities across the US and territories and provided care for 6.72 million unique Veterans in the fiscal year 2023, comprising the largest integrated healthcare system in the nation [17]. The VHA is the largest single provider of HIV care in the US, with over 31,000 Veterans living with HIV. This, along with its high proportion of male Veterans, makes it a potentially high-risk population for LGV. We aimed to analyze the occurrence of clinically defined LGV in a cohort of HIV-infected individuals cared for in VHA with positive CT test results to determine factors associated with LGV and its treatment. We are not aware of previous studies of LGV among US Veterans.

## 2. Materials and Methods

### 2.1. Data Sources and Study Cohort

This retrospective cohort study was performed on Veterans living with HIV and receiving VHA care between January 2016 and December 2023. The cohort of HIV-positive patients was obtained from the VHA Corporate Data Warehouse (CDW), a repository of VHA electronic health records (EHR) [18]. The CDW was queried to return individuals defined as being possibly HIV-infected, i.e., those meeting one or more of the following criteria: (1) the presence of HIV viral markers (antibody test with positive confirmatory testing or HIV-1 viral load) and (2) HIV ICD-10-coded outpatient or inpatient visit or problem list entry. Individuals were excluded if eligibility criteria indicated they were not Veterans or if they were deceased during the study period. We then compiled a list of patients with HIV infection who had also been infected with CT during the study period based on positive RT-PCR results. Equivocal and indeterminate CT RT-PCR results were not included. Using this cohort of HIV-positive/CT-positive Veterans, we performed a chart review of clinical progress notes from the time of CT positivity. LGV infection was considered present if one or more of the following CDC/CSTE clinical criteria were met [1,12]:Presence of proctitis as evidenced by rectal bleeding, mucoid, and/or hemopurulent rectal discharge, abdominal pain, anal pain, pain when passing stools, constipation, and/or tenesmus. In later stages, rectal fibrosis and/or strictures and anal fistulae;Presence of inguinal, femoral, or anorectal lymphadenopathy or generalized lymphedema of genital organs;Genital, oral, or rectal ulcerations or papules.

We determined that a patient with a positive CT sample met confirmed LGV criteria if a molecular test was performed confirming the presence of L1, L2, or L3 serovars, regardless of clinical symptoms. A probable case met one or more clinical criteria, and no other STI was present at the time of CT positivity, while a possible case was defined as one that met one or more clinical criteria for LGV. However, a concurrent STI was present, which could potentially cause LGV-like symptoms (e.g., gonorrhea (GC), syphilis, herpes simplex virus (HSV), mpox, and *Shigella*) was determined via chart review of laboratory results.

### 2.2. Analysis Methods

We compared HIV patients whose CT infection we determined met LGV clinical criteria to those whose CT infection did not meet LGV criteria for the following characteristics: age at CT diagnosis, birth sex, sexual orientation, race/ethnicity, rurality of patient address, US Census region of VHA facility state where diagnosis was made [19], period of military service [20], coinfection with other STIs, and CD4 count in cells/µL on date nearest to positive CT test.

For LGV clinical criteria-positive individuals, we analyzed additional characteristics such as the classification of LGV (confirmed, probable, or possible), presenting clinical syndrome, body site of positive CT sample(s), hospitalization, length of hospitalization, and antimicrobial treatment regimen. Individual patient treatment regimens were compared with the standard treatment regimen recommended by CDC [1]. Patients were assessed for adequacy of treatment regimen, treatment delays, and follow-up CT testing. Individuals receiving standard treatment for LGV were compared to those receiving non-standard treatment regimens (or no treatment) based on timing of treatment, time between CT testing and result completion, hospitalization, length of stay, follow-up CT testing at 3–12 months, sample site, classification of LGV (probable versus possible), and presenting clinical syndrome. These analyses were performed using chi-square or Fisher’s exact test for categorical variables and Mann–Whitney Wilcox test for continuous variables. All statistical analyses were performed using R version 4.3.0 (R Foundation for Statistical Computing, Vienna, Austria. www.R-project.org, accessed on 11 June 2024). A *p* value < 0.05 was considered statistically significant.

To evaluate the utility and accuracy of using ICD-coded encounter data (outpatient and inpatient) to identify LGV clinical criteria-positive cases, we calculated the sensitivity, specificity, positive predictive value (PPV), and negative predictive value (NPV) of A55 ICD-10 code, Chlamydial lymphogranuloma (venereum), using LGV clinical criteria described above as the “reference standard”.

### 2.3. Ethics Statement and Disclaimer

Data utilized in this study were obtained for the purpose of Public Health Operations in the VHA. No additional analyses outside of public health operational activities were performed. Therefore, this study was deemed to meet the requirements of public health surveillance as defined in 45 Code of Federal Regulations 46.102(I)(2). This project was approved by the Stanford Institutional Review Board (Protocol 47191, “Public Health Surveillance in the Department of Veterans Affairs”), and written informed consent was waived. The opinions expressed in this paper are those of the authors and do not necessarily reflect the position or policy of the US Department of Veterans Affairs or the US government.

## 3. Results

### 3.1. Characteristics of LGV Clinical Criteria-Positive Cases

From a total of 1381 HIV-positive patients reviewed who had a positive CT sample from any site, EHR revealed that 284 (20.6%) met clinical criteria for LGV; 179/284 (63%) were probable cases, and 105/284 (37%) were possible cases, with concurrent other STI present (Figure 1). No individuals had molecular testing for CT L1, L2, or L3 serovars; therefore, none met the confirmed LGV case definition. Clinical signs and symptoms of LGV presenting at diagnosis included proctitis in 230 cases (81%), compared to 71 (25%) with genital, oral, or rectal ulcers and 57 (20.1%) with inguinal, femoral, or anorectal lymphadenopathy, and 66 (23.2%) patients with more than one clinical symptom of LGV present (Table 1). CT was positive from rectal samples in 234 of 284 (82%) LGV clinical criteria-positive cases, 48 (16.9%) from urine, 12 (4.2%) from the pharynx, 3 (1.1%) from genital, and 1 (0.4%) from the cervix. In 14 cases (4.9%), CT was present in more than one sample site (rectum/pharynx: 9 cases, rectum/urine: 4 cases, and rectum/cervix: 1 case). The average number of clinically defined cases of LGV per year was 35.5. Cases peaked in 2017, then gradually decreased to their lowest occurrence in 2020, and increased again in 2021 (Figure 2). Cases occurred in Veterans in 38 states and in the District of Columbia and Puerto Rico, with over half of all cases occurring in 3 states: California (64 cases; 22.5%), Georgia (57; 20.1%), and Texas (39; 13.7%).

A total of 43 (15%) LGV clinical criteria-positive cases were hospitalized with a median age of 42 years (IQR 32.5–49). Thirty-seven (37/43; 86%) cases had proctitis as their presenting LGV syndrome, and 38 (88.0%) were CT-positive from a rectal sample site. Admitting diagnoses included anal fissure, perirectal abscess, rectal bleeding, fever, and severe pain. The median length of stay for hospitalized LGV clinical criteria-positive cases was 2.5 days (IQR 2–4.75).

### 3.2. Risk Factors for LGV

In comparing HIV-positive, CT-positive individuals who were LGV clinical criteria positive to those LGV criteria negative (Table 2), we found that the primary risk factors predisposing individuals to LGV were male birth sex (*p* = 0.004), MSM sexual orientation (including those identifying as bisexual and transgender) (*p* < 0.001), and presence of other STIs such as HSV, mpox, or *Shigella* (*p* = 0.011), but not GC or syphilis. The following factors had no significant correlation with LGV: age group, race/ethnicity, rurality of residence, US Census region of testing facility, period of military service, concurrent infection with GC or syphilis, and CD4 count.

### 3.3. Comparison of Standard vs. Non-Standard Treatment Regimens and Follow-Up for LGV

In Table 3, we compare characteristics of 124/284 (43.7%) clinically defined LGV cases who received standard treatment regimens (defined as receiving either doxycycline 100 mg orally two times/day for 21 days or azithromycin 1 g orally once weekly for 3 weeks; no patients received erythromycin) to 160/284 (56.3%) LGV clinical criteria-positive cases who received non-standard treatment (including four cases who received no documented treatment). All treatment regimens and numbers of patients receiving each type are described in Table 4. Of the four LGV clinical criteria-positive cases who received no treatment, three were unable to be reached by the provider to return for treatment; one had “CT negative” entered in the chart by the provider and was treated for GC but not CT.

We determined that the only factor significantly associated with receiving standard treatment compared to non-standard treatment was classification as probable LGV compared to possible LGV, with other STIs present (*p* = 0.003). Sample site, presenting clinical syndrome, hospitalization, time between CT testing and result completion, timing of treatment initiation, and presence or absence of CT follow-up testing at the recommended interval of 3–12 months post-positive CT test had no effect on whether patients fell into the standard treatment group or not. Patients who received non-standard treatment were not found to have a higher proportion of positive follow-up CT testing in our sample.

### 3.4. Utility of ICD-10-Coded Encounters for Identification of LGV Cases

ICD-10 LGV (A55) encounter codes were correctly applied to 32 out of the total 284 meeting LGV clinical case criteria (sensitivity 11.3%), while 1085 cases not meeting LGV clinical criteria of the total 1097 were not ICD-10-coded as LGV (specificity 98.9%). The PPV of ICD-10 code A55 in detecting LGV clinical criteria-positive cases was 72.7%, and the NPV was 81.2%.

## 4. Discussion

Between 2016 and 2023, we show that 20.6% of CT cases met LGV clinical criteria among a cohort of HIV-positive individuals, indicating that LGV occurrence among this Veteran population is likely underestimated. Fifteen percent of the LGV clinical criteria-positive cases we identified were hospitalized due to the severity of their anorectal and/or systemic symptoms, reinforcing the seriousness of LGV infection. Of great concern was our finding that less than half of the cases meeting LGV clinical criteria received standard recommended treatment regimens. Lack of recognition of LGV may hinder provider awareness and adequate treatment of cases, leading to complications and potential further transmission of this debilitating infection. Additionally, more than a quarter of cases did not have follow-up testing within the recommended time frame of 3–12 months, a finding reflected in a previous VHA study of CT and GC follow-up testing trends in VHA [21]. Our finding that LGV clinical criteria-positive cases who were diagnosed with other STIs (e.g., those deemed possible LGV by clinical criteria) were less likely to receive standard treatment than those without concurrent STI (deemed probable cases) was revealing. Providers may have felt that the presence of other STIs was the likely infectious cause of presenting signs and symptoms despite clinical evidence of LGV, such as proctitis or lymphadenitis.

Our detection of MSM as a primary risk factor for LGV is consistent with the findings of other studies [2,5,8,22], reinforcing the need for providers to be aware of LGV as a concern in this patient population, and for providers to obtain a detailed sexual history to ensure all appropriate anatomic sites are sampled for STI screening [23]. We also found that male birth sex was a risk factor for LGV; however, this may have been partly due to the high prevalence of MSM in our cohort. An interesting finding was the association of concurrent infection with STIs such as HSV, mpox, and *Shigella* (considered in this case to be an STI due to its manner of transmission via anal sex), but not with more common STIs, in particular GC or syphilis. It is unclear why these less common STIs were more closely linked to LGV, and this question may warrant further exploration.

Awareness of LGV among healthcare providers could be improved if a US Food and Drug Administration (FDA)-approved LGV RT-PCR laboratory test was made available in the US [7,16,24]. Without such a test, recent CSTE recommended changes to report guidance will be difficult to carry out as they will need to be based on subjective clinical criteria rather than objective laboratory criteria, as demonstrated by comparatively low LGV reporting in European countries where LGV laboratory diagnostics have not been implemented [16]. Nevertheless, providers should be educated regarding the presenting symptoms of LGV, high-risk patient populations, and the importance of prescribing adequate treatment for LGV, as the commonly prescribed 7-day doxycycline standard treatment for CT infection is regarded as inadequate for LGV and needs to be extended to 21 days when LGV is present. The possibility of LGV occurring in conjunction with other STIs should also be emphasized so that providers are aware that treatment may need to include coverage for more than one infectious agent and that standard GC/CT coverage is not adequate for LGV. Conversely, patients diagnosed with LGV should also be screened for other STIs, notable syphilis, GC, and HIV. Of note, as of June 2024, CDC guidelines recommend that MSM and transgender women diagnosed in the past 12 months with bacterial STIs (specifically syphilis, CT, or GC) should receive counseling on the benefits of self-administered doxycycline postexposure prophylaxis (doxy PEP) to prevent future infections with these agents [25].

Our analysis of LGV ICD-10-coded data highlights the obstacles presented to healthcare systems that may wish to assess the burden of LGV through the utilization of ICD-10-coded insurance claims rather than clinical data. These efforts could be hampered by the low sensitivity of the A55 ICD-10 code despite relatively high specificity. As has been demonstrated by other studies [26], researchers should proceed with caution when attempting to carry out public health surveillance based solely on coded data.

Our study was subject to several limitations. The study cohort was limited to Veterans with HIV, and therefore, findings may not be generalizable outside of this risk group. Expanding to a wider, non-HIV-infected population in a future study may be beneficial. In Europe, since 2019, the proportion of LGV cases reported has been increasing among HIV-negative MSM, largely driven by changes in clinical guidelines and more comprehensive testing strategies. In building our cohort of individuals with HIV, we did not include individuals who were treated with an HIV antiretroviral but who did not have evidence in the EHR of positive HIV laboratory testing; thus, we may have excluded individuals testing positive for HIV outside VHA who were treated and cared for within VHA. Our inclusion of Veterans ICD-10-coded for HIV in medical visits or in problem lists may have inadvertently included individuals miscoded as HIV who did not, in fact, have HIV infection. However, in general, our review of EHR notes identified these miscoded cases, and these patients were excluded from the cohort. Although nonconfirmatory LGV serologic testing is performed at some VHA facilities, as well as minimal use of laboratory-developed test (LDT) LGV RT-PCR tests not approved by the US FDA, the lack of availability of confirmatory CT L1-L3 RT-PCR testing in our cohort required us to rely on available clinical data utilizing CDC/CSTE-derived clinical criteria to classify CT cases as LGV. These criteria have not been compared to “gold standard” RT-PCR testing to establish sensitivity and specificity, and therefore, LGV cases may have been misclassified. Despite our reliance on clinical criteria only, our results are consistent with previously published work that used RT-PCR testing to confirm between 14 and 25% LGV infection among CT-positive samples [7,16]. Not all individuals in the cohort were tested for all STIs, so our assessment of concurrent STIs may be incomplete. We did not use the provider’s decision to treat for LGV as a proxy for meeting clinical criteria for LGV, and therefore, did not assess treatment among the 1097 patients whose CT infection did not meet clinical criteria for LGV. It is possible, therefore, that we could have missed LGV cases treated with longer duration of antimicrobials but for whom documentation of symptoms was lacking. Missing demographic data were problematic in our study and may have affected outcomes, specifically with respect to results pertaining to race/ethnicity and sexual orientation.

## 5. Conclusions

We report over 20% of CT cases met clinical criteria for LGV among HIV-infected US Veterans and that less than half of these individuals received recommended treatment regimens, indicating that LGV is likely underestimated and inadequately treated among this population. The availability of first-line CT testing to recognize serovars L1-L3 would improve providers’ ability to recognize and treat this serious infection. LGV public health messaging intended to raise awareness of symptoms, reduce stigma, and address equity should be targeted toward healthcare providers and individuals at high risk for infection. Healthcare providers should also be educated regarding populations most at risk for LGV infection and the importance of adequate treatment and follow-up of cases.

## Figures and Tables

**Figure 1 microorganisms-12-01327-f001:**
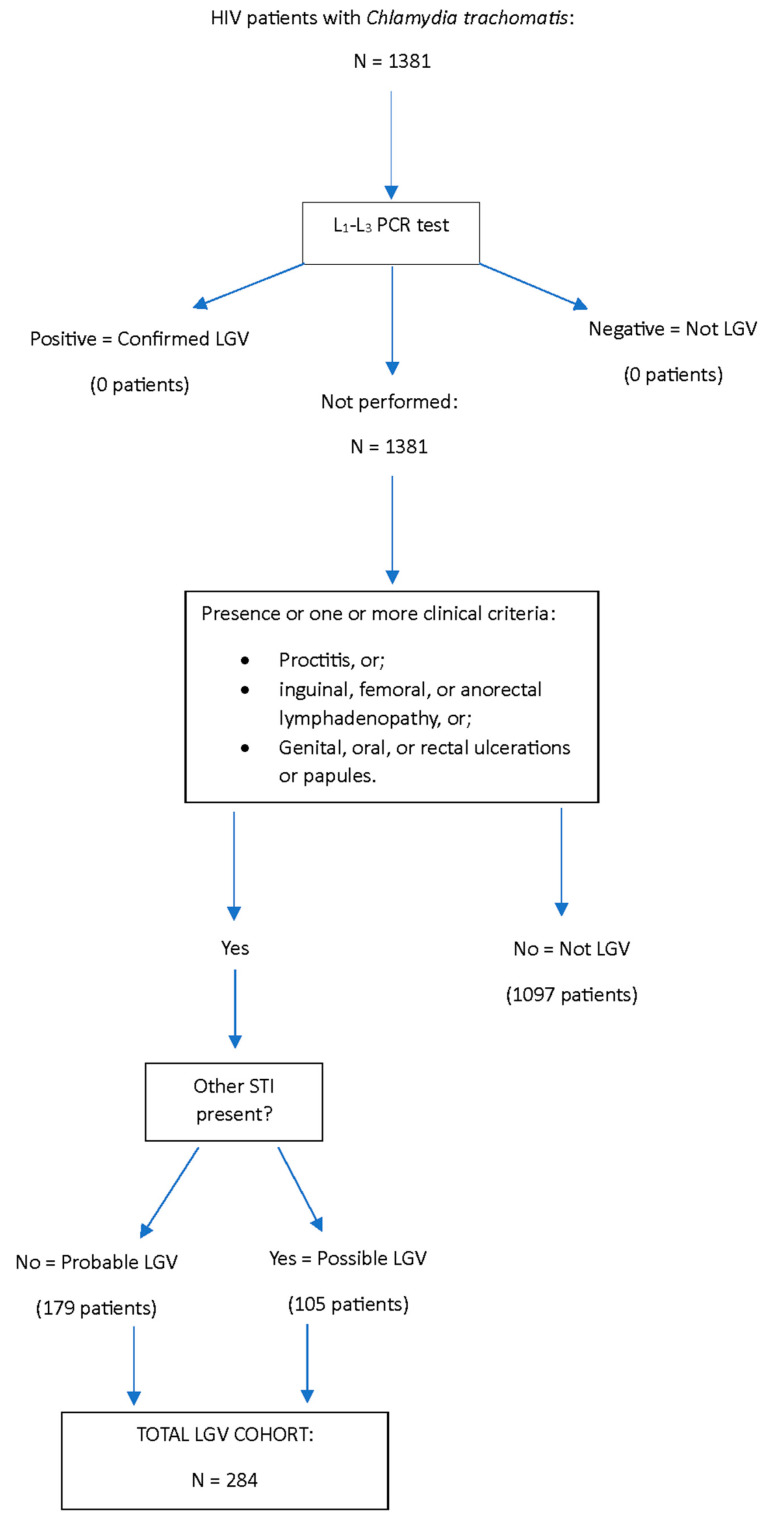
LGV clinical criteria positive case classification flow diagram, Veterans Health Administration, 2016–2023.

**Figure 2 microorganisms-12-01327-f002:**
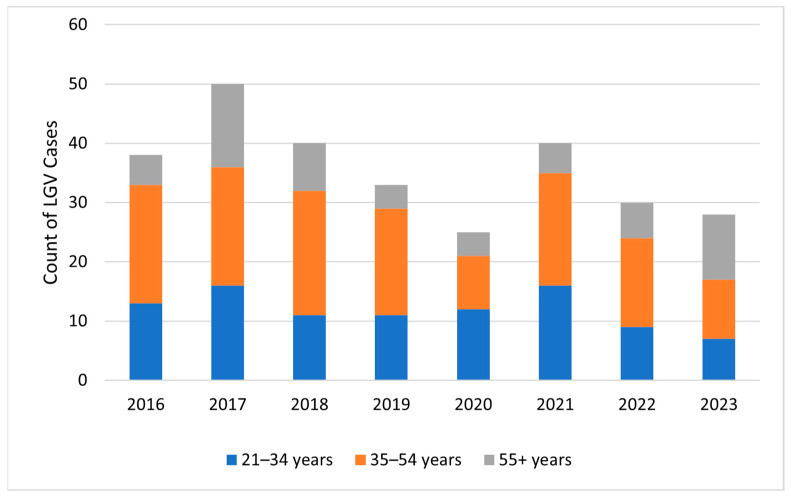
Clinically defined LGV cases by age group and year among HIV-infected individuals, Veterans Health Administration, 2016–2023. N = 284.

**Table 1 microorganisms-12-01327-t001:** Characteristics of clinically defined lymphogranuloma venereum (LGV) cases in HIV-positive individuals, Veterans Health Administration, January 2016–December 2023. N = 284.

Characteristic	Number (%)
Classification of LGV	
Probable	179 (63.0)
Possible (other STIs present)	105 (37.0)
Presenting clinical syndrome ^a^	
Proctitis	230 (81.0)
Genital, oral, or rectal ulcers	71 (25.0)
Lymphadenopathy (inguinal, femoral, or anorectal)	66 (23.2)
Sample site ^b^	
Rectal	234 (82.0)
Urine	48 (16.9)
Pharynx	12 (4.2)
Genital	3 (1.1)
Cervix	1 (0.4)

^a^ Sixty-six patients had more than one clinical symptom of LGV present. ^b^ CT was present in more than one sample site in 14 (4.9%) cases.

**Table 2 microorganisms-12-01327-t002:** Comparison of LGV clinical criteria-positive to LGV criteria-negative individuals. N = 1381.

	LGV Clinical Criteria Positive (n = 284)	LGV Clinical Criteria Negative (n = 1097)	
	N (%)	N (%)	*p*-Value ^a^
**Characteristic**			
Age Tested (Median (IQR))	42 (33–53)	42 (33–54)	0.694 ^b^
**Age Group**			
21–34 years	96 (33.5)	340 (31.0)	0.322
35–54 years	132 (46.5)	491 (44.8)	
55 years or older	57 (20.1)	266 (24.2)	
**Birth Sex**			
Male	284 (100.0)	1066 (97.2)	**0.004**
Female	0 (0.0)	31 (2.8)	
**Sexual Orientation**			
Bisexual/MSM/Transgender	241 (84.9)	670 (61.1)	**<0.001**
Heterosexual	18 (6.3)	224 (20.4)	
Unknown/Not defined	25 (8.8)	203 (18.5)	
**Race/Ethnicity**			
Non-Hispanic Black	157 (55.3)	613 (55.9)	0.311
Non-Hispanic White	87 (30.6)	281 (25.6)	
Hispanic/Latino	23 (8.1)	123 (11.2)	
Other Race/Ethnicity ^c^	6 (2.1)	27 (2.5)	
Unknown	11 (3.9)	53 (4.8)	
**Rurality of Residence**			
Urban residence	264 (93.0)	992 (90.4)	0.186
Rural residence	20 (7.0)	105 (9.6)	
**Census Region of Testing Facility**			
Northeastern	20 (7.0)	68 (6.2)	0.136 ^d^
Midwestern	23 (8.1)	141 (12.9)	
Southern	154 (54.2)	599 (54.6)	
Western	87 (30.6)	288 (26.3)	
Puerto Rico	0 (0.0)	2 (0.2)	
**Period of Military Service ^e^**			
Persian Gulf War	226 (79.6)	845 (77.0)	0.637 ^d^
Vietnam Era	58 (20.4)	250 (22.8)	
Korean Conflict	0 (0.0)	2 (0.2)	
**Concurrent STI ^f^**			
GC positive	61 (21.5)	240 (21.9)	0.885
Syphilis positive	53 (18.7)	186 (17.0)	0.498
Other STI positive ^g^	12 (4.2)	19 (1.7)	**0.011**
**CD4 count, cells/µL ^h^ (Median (IQR))**	627 (454–819)	642 (447–867)	0.442

^a^ Chi-square test except where indicated. ^b^ Mann–Whitney Wilcox test. ^c^ Other race/ethnicity = non-Hispanic American Indian/Alaskan Native, or non-Hispanic Asian or non-Hispanic Native Hawaiian/Other Pacific Islander. ^d^ Fisher’s exact test. ^e^ Korean Conflict: 6/27/1950–1/31/1955, Vietnam Era: 1/1/1955–5/7/1975, Persian Gulf War: 8/2/1990-current. ^f^ Not all patients were tested for all STIs. ^g^ Other STIs = HSV, Mpox, and Shigella. ^h^ Most recent CD4 count prior to positive CT test.

**Table 3 microorganisms-12-01327-t003:** Comparison of clinically defined LGV cases receiving standard treatment ^a^ to those receiving non-standard ^b^ (or no ^c^) treatment. N = 284.

	Standard Treatment ^a^ (n = 124)	Non-Standard Treatment (n = 160)	
	N (%)	N (%)	*p*-Value ^d^
**Characteristic**			
**Classification of LGV**			
Probable	90 (72.6)	89 (55.6)	**0.003**
Possible (other STIs present)	34 (27.4)	71 (44.4)	
**Sample Type**			
Rectal	108 (87.1)	126 (78.8)	0.067
Nonrectal	16 (12.9)	34 (21.3)	
**Presenting Clinical Syndrome**			
Proctitis	106 (85.5)	124 (77.5)	0.089
Genital, oral, or rectal ulcers	29 (23.4)	42 (26.3)	0.581
Lymphadenopathy (inguinal, femoral, or anorectal)	30 (24.2)	27 (16.9)	0.127
**Hospitalization**			
Yes	23 (18.5)	140 (12.5)	0.158
No	101 (81.5)	20 (87.5)	
**Median Length of Stay (days)**	3 (IQR 2–6)	2 (IQR 1–4)	0.162
**Time Between CT Testing and Result Completion**			
0–3 days	59 (47.6)	87 (54.4)	0.381
4–7 days	57 (46.0)	67 (41.9)	
>7 days	8 (6.5)	6 (3.8)	
**Timing of Treatment Initiation**			
Within ≤7 days of + CT test	111 (89.5)	146 (91.3)	0.621
>7 days of + CT test	13 (10.5)	14 (8.8)	
**CT Follow-up Testing Within 3–12 months of + CT Test**			
Yes	88 (71.0)	122 (76.3)	0.315
No	36 (29.0)	38 (23.8)	
**CT Follow-up Test Result (n = 210 tested)**			
Positive	6 (6.8)	16 (13.1)	0.207
Negative	82 (93.2)	106 (86.9)	

^a^ Standard treatment group: those treated with doxycycline 100 mg orally 2 times/day for 21 days or Azithromycin 1 g orally once weekly for 3 weeks. ^b^ Non-standard treatment group: includes any Doxycycline or Azithromycin regimen not meeting dosage or duration requirements of standard treatment. Also includes 2 patients treated with minocycline. ^c^ Not treated: of 4 LGV clinical criteria-positive cases who received no documented treatment, 3 were unable to be reached by the provider to return for treatment; 1 had “CT negative” entered in the chart by the provider and was treated for GC, but not CT. NOTE: all 4 were included in the “non-standard treatment” group for analysis purposes. ^d^ Chi-square for categorical and Mann–Whitney Wilcox for continuous variables.

**Table 4 microorganisms-12-01327-t004:** Treatment regimens among clinically defined LGV cases. N = 284.

**Standard Treatment Regimens: (n = 124)**	**No. (%)**
Doxycycline 100 mg orally 2 times/day for 21 days	121 (97.6)
Azithromycin 1 g orally once weekly for 3 weeks	3 (2.4)
**Non-standard treatment regimens: (n = 160)**	
Doxycycline 100 mg orally 2 times/day for 7 days	55 (34.4)
Doxycycline 100 mg orally 2 times/day for >7 days but <21 days	19 (11.9)
Minocycline 100 mg orally 2 times/day for 21 days	2 (1.3)
Azithromycin 1 g orally one time	78 (48.8)
Azithromycin 500 mg orally 2 times/day for 5 days	2 (1.3)
No treatment *	4 (2.5)

* Of 4 LGV clinical criteria-positive cases who received no treatment, 3 were unable to be reached by the provider to return for treatment; 1 had “CT negative” entered in the chart by the provider and was treated for GC, but not CT.

## Data Availability

The data that support the findings of this study are available from the corresponding author upon reasonable request.

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
