# Peer review of "Clinically Defined Lymphogranuloma Venereum among US Veterans with Human Immunodeficiency Virus, 2016–2023"

_microorganisms, 2024, doi:10.3390/microorganisms12071327_

Round 1

Reviewer 1 Report

Comments and Suggestions for Authors

Oda and colleagues have provided an approach on the estimation of lymphogranuloma venereum (LGV) prevalence among HIV-positive US veterans. The approach is interesting, especially considering the described lack of serovar-specific diagnostic assays. I have a few recommendations on how the manuscript can be further improved.

1.) Title: It should be made clear in the title that diagnoses were based upon clinical criteria rather than LGV-specific PCR. Otherwise, readers might erroneously think that the article is on PCR-confirmed cases.

2.) Introduction, lines 61-67: The authors themselves state that longer therapy is necessary to achieve therapeutic success in case of LGV compared to serovar D-K infections. It is not entirely clear why the authors did not more thoroughly make use of therapy-control-assessments to verify or falsify their assumption of relatively high LGV-prevalence within the assessed population at risk. At least, this aspect should be mentioned in the limitations section.

3.) Methods, line 94-100: Do the authors have diagnostic accuracy estimations (in terms of sensitivity and specificity) for their case definitions? At least from the literature? Such information would be very helpful for the estimation of the validity of their case definition-based approach. If such information is not available and also cannot be estimated with acceptable effort by the authors, this should at least be stated as a limitation of this work in the limitations section.

4.) Results chapter, general comment: In the results chapter, clinically defined likely LGV cases are generally treated as more or less certain LGV cases. Slight rewording might be helpful to underline the uncertainty associated with the study’s premises.

5.) The proportions of applied therapeutic regiments is interesting; I recommend for presenting the respective table in the manuscript’s main text and not just in the supplementary materials.

Reviewer 2 Report

Comments and Suggestions for Authors

In general the manuscript is interesting in the prevalence of lymphogranuloma venereum (LGV) in a specific cohort of patients in the US. It is well-redacted and written, summarised clinical and patient data providing a statistical correlation between them.

Some important questions or recommendations are provided below to the authors:

Could you provide some information about the treatment problems of CT infections? exist antimicrobial resistance in CT?  Is it possible to explain more clearly about ICD-10 code?

Minor comments:

In line 58, add a period after "et al" and write in italics, please.

Finally, it is a pity that PCR did not exist for the identification of L1-L3; this could improve the present manuscript largely. We hope that the authors provide a future manuscript with the application of this laboratory technique in order to present more consistent results.
